# Understanding Elderly Chinese Cancer Patients in a Multicultural Clinical Setting: Embracing Mortality and Addressing Misperceptions of Vulnerability

Yvonne W. Leung [1,2,*], Enid W. Y. Kwong [3], Karen Lok Yi Wong [4], Jeremiah So [1], Frankie Poon [1], Terry Cheng [1,5], Eric Chen [6], Alex Molasiotis [3] and Doris Howell [5]

1.  Department of Psychiatry, University of Toronto, Toronto, ON M5S 1A1, Canada
2.  College of Professional Studies, Northeastern University, Boston, MA 02115, USA
3.  School of Nursing, Hong Kong Polytechnic University, Kowloon, Hong Kong
4.  School of Social Work, University of British Columbia, Vancouver, BC V6T 1Z4, Canada
5.  Department of Supportive Care, Princess Margaret Cancer Centre, University Health Network, Toronto, ON M5G 2C4, Canada
6.  Department of Medical Oncology, Princess Margaret Cancer Centre, University Health Network, Toronto, ON M5G 2C4, Canada
*   Correspondence: yw.leung@utoronto.ca

**Abstract:** Chinese patients face higher risks of gastrointestinal (GI) cancers and greater cancer-related deaths than Canadian-born patients. The older population encounters barriers to quality healthcare, impacting their well-being and survival. Previous studies highlighted Chinese immigrant perceptions of not requiring healthcare support. During the COVID-19 pandemic, their underutilization of healthcare services garnered attention. The present study explores the experiences of older Chinese cancer patients to improve culturally sensitive cancer care. A total of twenty interviews carried out in Cantonese and Mandarin were conducted with Chinese immigrants, aged 60 or above, diagnosed with Stage 3 or 4 GI cancer. These interviews were transcribed verbatim, translated, and subjected to qualitative descriptive analysis. Among older Chinese immigrant patients, a phenomenon termed "Premature Acceptance: Normalizing Death and Dying" was observed. This involved four key themes: 1. acceptance and letting go, 2. family first, 3. self-sufficiency, and 4. barriers to supportive care. Participants displayed an early acceptance of their own mortality, prioritizing family prosperity over their own quality of life. Older Chinese patients normalize the reality of facing death amidst cancer. They adopt a pragmatic outlook, acknowledging life-saving treatments while willingly sacrificing their own support needs to ease family burdens. Efforts to enhance health literacy require culturally sensitive programs tailored to address language barriers and differing values among this population. A strengths-based approach emphasizing family support and practical aspects of care may help build resilience and improve symptom management, thereby enhancing their engagement with healthcare services.

**Keywords:** Chinese immigrant; supportive care; psychosocial needs; culture; barriers; cancer

## 1. Introduction

By the year 2030, primarily due to the aging population, there will be more seniors than children in North America. The ethnic Chinese group is one of the largest visible ethnocultural minorities in North America [1] with approximately 18.8% of this population over the age of 65. The underutilization of healthcare services among seniors from ethnic minority backgrounds is a significant concern [2,3]. This issue has been heightened by the COVID-19 pandemic, leading to a notable decrease in the use of vital healthcare services like cancer screening among this group [4–6]. This disproportionate effect highlights the need for increased attention and targeted support to ensure equitable access to essential

healthcare services for these communities. Chinese immigrants encounter challenges in accessing quality healthcare services [7] due to language barriers, low income, and limited health literacy [8], which have adverse effects on timely diagnosis, overall survival rates, quality of life, and healthcare costs [9]. Chinese immigrants are at an elevated risk of developing gastrointestinal (GI), nasopharynx, and lung cancers when compared to the Caucasian population [10,11]. Evidence also pointed to a lack of cancer screening in the Chinese population due to low health literacy [12,13]. The existing literature has primarily focused on assessing vulnerability and identifying the needs assessment of this population, as they necessitate additional support and specialized attention [14]. For example, the literature addressing the healthcare of Chinese immigrants often focuses on their vulnerability related to their socioeconomic status and cultural factors [15]. The literature has indicated that Chinese immigrants tend to exhibit poor health literacy. However, when provided with recommendations, they displayed more favourable attitudes toward health checkups and perceived fewer obstacles to mammographic screening [8].

Chinese immigrants commonly prioritize family over themselves, embracing cultural norms and holding misconceptions about Western medicine [16]. Chinese culture traditionally revolves around the construct of collectivism rather than individualism [17]. Collectivists value harmony and group goals over oneself [17]. Chinese culture also places a strong emphasis on not burdening society, which can act as a barrier preventing Chinese immigrants from accessing healthcare services [18]. Many older Chinese patients depend on their adult children for primary care, refraining from seeking additional help for non-medical appointments, often driven by their self-sacrificing nature [19]. Many also hold the belief that the primary responsibility for one's health lies with the individual rather than relying on the healthcare system. Some Chinese adults expressed the viewpoint that a healthy body should possess the inherent capability to fight off diseases on its own [20]. This perspective can lead to delays in seeking appropriate care until symptoms reach an unbearable level.

Chinese immigrants, as well as their family caregivers, often face language barriers and communication difficulties when interacting with healthcare providers. In particular, language barriers hinder Chinese patients from comprehending pertinent information. Consequently, they encounter difficulties in effectively conveying their needs and preferences [8,21], including their interest in utilizing traditional Chinese medicine [22], and dealing with misunderstandings related to preventive medicine [16]. This, in turn, reinforces their beliefs regarding the antagonistic and potentially harmful attributes of Western medicine [16], the intrusiveness of blood tests, and the stigmatizing nature of illnesses [20,23]. In Chinese culture, death and dying are often regarded as a taboo subject, discouraging Chinese immigrants from engaging in discussion on these topics and impeding their communication with service providers, particularly on matters related to advance care planning [24]. Therefore, discussion topics of palliative care and end-of-life planning present a challenge for healthcare providers [25,26]. The underutilization of healthcare services among immigrants is compounded by the fear of becoming a burden to both their families and the broader society [27,28]. These complex sociocultural perspectives lead patients to disregard their own needs [29,30]. This viewpoint contributes to ill-informed treatment decisions, the pursuit of aggressive treatment near the end of life, and a greater frequency of emergency room visits [31,32].

To address the lack of understanding of the care needs of Chinese patients, the existing literature has primarily concentrated on factors such as language, education, and socioeconomic vulnerability [33,34]. Few studies offer culturally relevant insights into the factors contributing to the low healthcare participation among Chinese immigrants. As such, the primary objective of the present study was to investigate the experiences of older Chinese immigrants with cancer within a culturally diverse metropolitan area, with a particular focus on their cultural perspectives. A sociocultural perspective was adopted as the theoretical framework to understand the lived experiences of older Chinese individuals, highlighting the complex interrelationship among various viewpoints related

to disease and death. Traditional Chinese culture certainly embodies a hierarchical and collectivist structure marked by a harmonious balance of duty, obedience, and proclivity to avoid conflicts [35]. Within the context of collectivism, Chinese virtues adopt a pragmatic approach that involves acceptance and adaptability to change, all the while prioritizing the preservation of harmonious relationships in the face of navigating through crisis [36]. These perspectives will be incorporated into our context when investigating the implications for their access to supportive services in cancer.

## 2. Materials and Methods

Research Design: This research is an exploratory, qualitative study based on a qualitative descriptive methodology [37]. The research team consisted of two nursing associate professors, a social work assistant professor, a researcher with a PhD, and 10 research volunteer staff who were proficient in Cantonese and/or Mandarin and English languages. Participants were interviewed according to a semi-structured, open-ended question guide about their experience with cancer, treatment, and self-management knowledge at the Princess Margaret Cancer Centre in Toronto, Canada. To ensure comprehensive data collection, participants were encouraged to openly share their feelings and thoughts during the interview process. This study was approved by the institutional research ethics board.

Sampling Method: A purposive sampling of 20 Chinese immigrants, aged 60 or older, with confirmed diagnosis of stages II to IV GI cancer were recruited between May 2014 and May 2015 to reach saturation. Patients were offered the option for either face-to-face or telephone interviews, conducted in either Chinese or English. All participants expressed a preference to be interviewed in Mandarin or Cantonese.

Recruitment Procedures: The daily GI clinic schedules were screened to identify individuals of Chinese descent based on their last names and age. Oncologists were informed about the study to aid in the recruitment process. After identifying potential participants, the attending physician first informed the patients about the study occurring in the GI clinic. Upon agreement, the interviewer (EK) would approach the participants and obtain signed consent. Face-to-face interviews were promptly scheduled after the clinic appointment, while telephone interviews were arranged on a different day at the participant's convenience. All interviews were digitally audiotaped, transcribed, and translated.

Data Analysis: All interviews were transcribed verbatim in Chinese and later translated into English by trained volunteer research assistants. To ensure accuracy, a backward translation was conducted for the initial four cases (YL). All transcriptions and translations were verified by a second research assistant volunteer. The research group led by the co-investigator (YL) met regularly to discuss the discrepancies and conflicts in translations of difficult concepts until consensus was reached through a combined open-axial coding process.

The data were analyzed following the procedures as outlined by the Qualitative Descriptive Method [38]. To preserve the integrity of each story, a summary including the unique aspects and cultural elements was outlined for each individual case. This involved extensive readings of each transcript to identify relevant concepts of interest. A conceptual map and R Qualitative Data Analysis (RQDA) software version 0.2-8 [39] were employed to aid in conceptual organization and the development of themes. As themes emerged, the researcher validated each theme by revisiting individual cases. Finally, a thematic profile was constructed to reflect the overall studied phenomenon.

## 3. Results

Semi-structured individual interviews were conducted with Chinese-speaking, Cantonese or Mandarin, first-generation immigrants. Out of 22 Chinese immigrant patients, a total of 20 patients (mean age = 66.5, SD = 6.3, range 60–84) diagnosed with stage II–IV GI cancer participated in the personal interviews. All the participants preferred to communicate in Chinese. A total of 65% of participants were unemployed (See Table 1). Three

participants underwent interviews in the company of a primary caregiver, such as a spouse or adult child, to help with communication.

**Table 1.** All characteristics of participants who completed personal interviews (*n* = 20; *n* (%)).

| | |
|---|---|
| Sex | |
| Male | 15 (75) |
| Age | |
| >70 | 6 (30) |
| 66–70 | 5 (25) |
| 60–65 | 9 (45) |
| Employment status | |
| Full time | 2 (10) |
| Part time | 3 (15) |
| Unemployed | 13 (65) |
| Retired | 1 (5) |
| Permanent disability | 1 (5) |
| Education | |
| Less than elementary | 1 (5) |
| Elementary | 5 (25) |
| High school | 5 (25) |
| University, college or higher | 9 (45) |
| Location of disease | |
| Liver | 6 (30) |
| Colorectal | 9 (45) |
| Pancreatic | 3 (15) |
| Other locations | 2 (10) |
| Stages of disease | 3 |
| II | 3 (15) |
| III | 2(10) |
| IV | 9 (45) |
| Unstated | 6 (30) |

The overarching theme that emerged was self-reliance as a virtue and four related subthemes including acceptance and relinquishment, family first, self-sufficiency, and barriers to supportive care (see Figure 1). The findings revealed that participants adopted a philosophical yet pragmatic perspective, recognizing that their lives were reaching their conclusions. They believed that allocating resources toward supportive care needs perceived as non-life threatening would potentially compromise family or societal resources. Each subtheme and its related subcategories are discussed in detail below.

*3.1. Acceptance and Relinquishment*

This subtheme explores the various perspectives and beliefs surrounding death and dying among elderly Chinese cancer patients. It encompasses several subcategories that describe their attitudes: cancer is a death sentence, death is a normal part of life, and quality over quantity of life. Most participants expressed a deep sense of acceptance, attributing it to the belief that life is predetermined and the future is preordained by a deity or higher power. They perceived altering life's trajectory as a contradiction to the natural order, akin to opposing nature's will. Participants described a readiness to confront death and dying, exhibiting a sense of preparedness for the end of life, both emotionally and spiritually. This readiness was informed by their cultural and religious beliefs, which emphasized acceptance and surrender to the natural order.

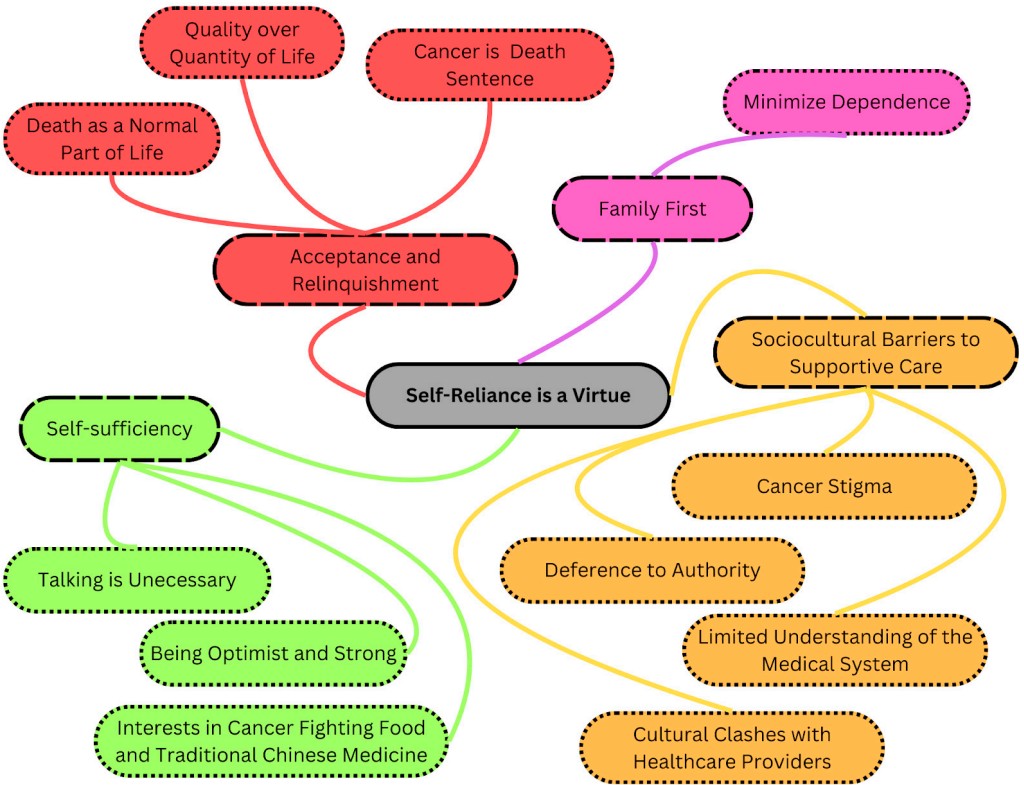

**Figure 1.** Summary of overarching theme, subthemes, and subcategories. Note. The border style corresponds to the grouping: solid border marking the overarching theme, dashed border marking the subthemes, and dotted border marking the subcategories.

### 3.1.1. Cancer Is a Death Sentence

Many participants regarded cancer as an inevitable death sentence, foreseeing imminent suffering caused by both the disease itself and the treatment involved. They viewed cancer as a terminal illness with little hope for recovery or long-term survival. Therefore, this perception influenced their acceptance of their prognosis and shaped their thinking that it is futile to even attempt to change the situation or cope with the disease:

> *"My disease is fatal. Even though everyone puts so much effort to help me with my treatments, including the government, hospital, doctors and you (EK) as well, but my disease is still fatal nevertheless... But I'm afraid that this disease cannot be cured, I could leave this world any time... I'm going to die no matter what. The treatment extends my life for a couple months. But I'm not sure if it is worth going through these painful experiences (treatments) just to extend my life for a couple extra months"*
> —022F.

### 3.1.2. Death as a Normal Part of Life

Participants commonly articulated the view that death is a natural and inevitable part of the life cycle. They accepted the idea that all living beings eventually face mortality and viewed their own impending death with a sense of resignation and philosophical acceptance. While acknowledging the inevitability of death and dying, some participants were resolute in moving forward and finding a deeper spiritual purpose in life. Others discussed embracing life as it unfolds, adopting a non-emotional and reactive attitude. The phrase, "I am not afraid", was reported by all the participants. Participants commonly expressed that excessively dwelling on or distressing about death and dying should not be a concern, given the inevitability of aging: *"This is a stage in life... I am not worried about death only if it is in a natural way... I did not ask and did not want to ask (the doctor about the life expectancy)."*—005M. A participant, whose cancer had metastasized to the lungs, expressed

a fatalistic view on death and dying: *"But what can you do?... That's it. . . I don't worry about things. If I die, I die. If that's it, that's it. If I die tomorrow, that's it"*—003M. Others exhibited a diminished emotional reaction towards death and dying:

> *"If I were to be sick, I'll let it be. If I were to die, so be it. No big deal. . . What can you do? Some things cannot be changed even if you want to. So you can only do it this way. . . It was useless saying anything. . . In this world, living and dying is a normal course of life. . . There is nothing to be optimistic or pessimistic about."*
> —016M.

Many participants anticipated the inevitable aspects of life and expressed contentment, noting that they had led a fulfilled and satisfying life. This was reflected in their sense of accomplishment with work fulfillment, grown-up children, and grandchildren:

> *"My feeling towards cancer was not so big of a reaction. I am already quite old now right? So it's normal to be sick. The kids are all grown up, all of them are working now. There is no financial problem. And I have lived a hard life for decades, and I have paid off my debts."*
> —014M.

Participants showed a sense of acceptance regarding death and dying. Many often mentioned the time of death was predetermined by a higher power or God. Death was viewed as a destination, whether or not they felt it was unnecessary to change the situation.

### 3.1.3. Quality over Quantity of Life

Rather than focusing solely on prolonging life at all costs, many participants emphasized the importance of quality of life. They made clear their preference for maintaining dignity, autonomy, and comfort, even if it meant accepting a shorter lifespan. They viewed ongoing treatment for advanced disease as a waste of societal resources. This perspective prioritized well-being and fulfillment over mere survival:

> *"My wife was allergic to chemotherapy, so she had to do this special chemotherapy instead, then she stopped eating entirely because she had lost her appetite. . . If treatment compromises my quality of life I'd rather give up treatment than live longer. I'd rather be walking than crawling (wasting away) . . . Alcohol harms the liver right? That means if I want to die earlier, I only need to drink more?"*

> *"Yes I'm going to stop the treatment because I'm going to die no matter what. The radiotherapy extends my life for a couple months, but I don't think it is worth to go through these painful experiences."*
> —019M.

Some expressed apprehension regarding the potential side effects of treatments that could significantly diminish their quality of life, leading them to outrightly dismiss such treatments:

> *"If I were to have cancer recurrence right now or anything alike, I would refuse treatment. The treatment would cause my quality of life to decrease significantly. There is no point living like that. . . I'd then choose to live in (palliative care) and pay a funeral home some money to bury myself somewhere."*
> —007M.

Participants generally equated cancer to death and accepted death as a healthy process of life. Therefore, they believed there was no need to insist on a desire to live or overthink the meaning or the cause of cancer.

### 3.2. Family First

Many participants expressed a strong value for family, de-emphasizing the self, and sacrificing personal needs as family well-being came before their own: *"My kids also asked*

*if I wanted anything. I told them I don't need anything. It is okay as long as they are safe and sound."*—009M. Family support was the major source of support for most participants:

> *"My biggest worry is my family worrying about me. . . I think it would help my family if I face it (my illness) optimistically. . . Of course family is the most important. Their caring helped the most. In fact many things in my daily life depend on my family."*
> —002F.

Minimize Dependence

Not only did participants minimize the burden placed on their families, but they also found practical ways to continue supporting their families:

> *"After all, there is mental stress and burden in my life. Two of my grandchildren, one from my son, another from my daughter, are autistic. . . So we would think of ways to lessen their (son and daughter) burdens. I could take care of myself just fine, so I could help them out a bit."*
> —011M.

Some mentioned not receiving much support in Canada, as their friends and relatives resided overseas. They expressed concerns about financially burdening their children, as they were no longer able to substantially contribute to the family. Further, they did not want their children taking time off to care for them, considering their own well-being as trivial compared to their children's livelihoods:

> *"Normally I don't want to distract my son-in-law from work. It is because he is working as an engineer who has lots of responsibilities. Our 8-people family relies on him as he is the only person who works. Due to my illness, I can't even do housework. We have four elderlies and two young children and him (patient's son-in-law) at home. . . I am trying my best to not get him involved. I don't want him to be involved with someone who is dying. If he lost his job, what should the family do? . . . My daughter told me that she wanted to stop going to school and stay with me during the treatment. But I cannot ruin her future because of a dying person like me."*
> —012F.

Participants consistently expressed the pivotal role of family as their source of strength and support throughout their experiences with cancer. As a result, they considered their own well-being relatively insignificant compared to the prosperity of their families. They were willing to self-sacrifice in various aspects to uphold family harmony.

*3.3. Self-Sufficiency*

Many participants placed importance on self-sufficiency, even in the absence of substantial social support. This was reflected in three subcategories: a belief that talking about their situation was unnecessary, a focus on staying optimistic and strong, and an interest in cancer-fighting foods and traditional Chinese medicine. They aimed to maintain their health through light exercise and healthy diets, intending to reduce the burden on others and society as a whole. Overall, participants expressed that maintaining independence in old age was a virtue that would benefit future generations and society.

3.3.1. Talking Is Unnecessary

Participants generally defined health and disease primarily from a physical standpoint, placing emphasis on emotional expression. Moreover, some believed that communicating their concerns to loved ones would not be beneficial for them:

> *"I don't think that I need to discuss it with others. There is no use talking about it. At the most it is a release; but after all they are just words."*
> —008M.

Certain participants held the belief that one should be cautious of limited resources and only utilize them when absolutely necessary:

*"I did not use it (the medical/ hospital support) because I did not think I needed it and I am not depressed. . . I am not greedy, not stressed, not worried. If I am worried then I will be checking here and there for information, how to prevent (prolong his life) . . . Not really interested (in the supportive care)."*

—005M.

They also expressed the view that the available supportive care services were not tailored to their specific needs, and the absence of transportation services posed an inconvenience:

*"Right now I have no need for such things (supportive care/community services) . . . It is very inconvenient for elderly to travel around if they can't drive anymore . . . I don't need others to help me, I am not used to that."*

—003M.

### 3.3.2. Being Optimistic and Strong

Many participants believed that maintaining optimism was crucial in battling the disease and preserving independence. They were confident in their ability to maintain positivity, viewing being downcast as a personal choice that one should actively avoid:

*"I was upset but I tried to have a positive attitude. This is a stage in life and if I can overcome it then I will overcome it. But I need to maintain a quality of life."*

—019M.

Certain participants opted to embrace optimism despite the unchangeable outcome:

*"People told me to be more optimistic and I thought 'oh yeah that's right'. Since it's the reality and being depressed will still lead to the same outcome so why not be more optimistic."*

—015M.

One participant acknowledged the difficulty in maintaining positivity due to family obligations:

*"I did not see a social worker, no time for these services because I need to make money to support my grandkids . . . I think I will be able to self-manage my mood. I listen to music and exercise a lot. I forced myself to be positive but sometimes I find it very difficult."*

—011M.

### 3.3.3. Interests in Cancer-Fighting Food and Traditional Chinese Medicine

When asked about the kind of services that older Chinese patients wanted, the majority indicated an increased need for supportive services of translators and financial assistance. A considerable number of participants expressed interest in learning about the kinds of food and Chinese medicine available that could strengthen their ability to fight cancer and aid in recovery from cancer treatments. Participants mentioned that their need for specific dietary considerations and traditional practices was often overlooked because hospital staff did not understand their importance to Chinese patients. Seeking active yet independent self-care behaviours, patients leaned towards Chinese medicine because of its natural properties, which aligned with their health philosophy.

One participant struggled to express himself, so his wife spoke on his behalf.

Wife of the patient: *"He (the patient) always wanted to ask the doctor about this. You know Chinese people often have those Chinese medicine, treatments, medications that can help with the cancer. . . Like now he is undergoing cancer treatment or after chemo are there any Chinese medicine to help recovery or would strengthen the body. . . He wants to know if there are any things like this and whether it is effective."*—006M.

Participants held a notably positive belief regarding traditional Chinese medicine:

*"Traditional Chinese medicine is simple, easy to do with no side effects. I have tried the 'sunbathe' before recommended by a Chinese doctor. It was quite effective."*

—018F.

Participants attributed part of the positive outcomes to the Chinese medicine they used:

*"I went to Vancouver to see this famous Chinese doctor who is known for his treatments. . . I was given Chinese medicine. . . I have continued taking it even when I was undergoing chemotherapy. . . I have felt better psychologically. . . No side effect and it has strengthened my body. The medicine was high in protein so it kind of saved my life because I was not eating enough."*

—002F.

### 3.4. Sociocultural Barriers to Supportive Care

Many participants experienced challenges related to language proficiency and health literacy, encompassing various social barriers to accessing healthcare. This subtheme comprises four subcategories: issues in the medical system, deference to authority, cultural clashes with healthcare providers, and cancer stigma. Some patients expressed that they were never asked about their psychosocial needs. Participants admitted that many illness-related concerns remained inadequately communicated or addressed by oncologists mainly due to language barriers. Some described cultural differences, particularly regarding the use of alternative medicine, which resulted in miscommunication. Moreover, many regarded themselves as having low literacy and were unaware of available resources for cancer management.

### 3.4.1. Limited Understanding of the Medical System

Certain participants appeared to possess limited knowledge about their disease or lacked an understanding of the limitations of diagnostic tools and the healthcare system:

*"In fact two years before I had done couple of ultrasounds but nothing was detected. I find this the most frustrating thing."*

—015M.

Participants expressed frustration regarding extended wait times for multiple procedures that needed to be carried out before treatments. One caregiver spoke on behalf of her father about such concern.

Patient's daughter: *"The discovery of my father's cancer and the diagnostic procedures took a long time. There was also a long wait time for the treatment preparation and process. I wondered if it was worth its wait in Canada or returning to China, that would be much faster but cost more."*—016M.

Some individuals found the Canadian healthcare system less efficient compared to healthcare systems in their home countries, where they could receive treatment almost immediately:

*"They (doctors) didn't need to do many check-ups in China, things like cardiogram, in order to operate on you. But here you would need to check many things, like cardiogram and EEG, to make sure your heart and lungs are good. It (the process for surgery) was a lot simpler in China."*

—011M.

### 3.4.2. Deference to Authority

Participants showed deference to authority due to limited medical knowledge. As a result, they placed complete trust in professionals with the highest qualifications when making life-saving decisions. Once participants trusted their oncologist, they refrained from questioning their oncologist's decisions or expressing reluctance to discuss concerns or alternative options for their conditions:

*"Because I don't think (psychosocial support) is necessary, so I didn't look for them. . . doctor's explanation is enough. I did ask my doctor about taking Chinese medicine but he told me not to take it because I was going to get chemotherapy and radiotherapy. . . I have heard about combining Chinese and Western medical treatments for cancer but I decided to listen to my doctor."*

—008M.

One responded mentioning disregarding advice from his Canadian doctor not to receive surgery, instead returning to China for a more radical treatment. This decision was influenced by the doctor's reputation, particularly their designation as a 'professor' and their experience with a larger case volume:

*"The (Canadian) doctor said that it's best not to remove it for my age. Radiotherapy was fine, this way the body wouldn't be too depleted. But back in China, they said it's best to remove it. Because I trust the doctors in China very much. They are all professors. Plus, they've seen thousands of cases a year."*

—010M.

### 3.4.3. Cultural Clashes with Healthcare Providers

Participants exhibited reluctance to discuss misunderstandings about their conditions or beliefs in traditional Chinese medicine. Many felt that cultural differences led to distinct beliefs and practices, creating a distance between them and healthcare providers. The daughter of a patient expressed concern about her worry of disapproval by her father's doctor regarding their interest in traditional Chinese medicine. The doctor cautioned them against using traditional therapies without thorough explanations.

Patient's daughter: *"I have thought of Chinese medicine and wonder if it would benefit Dad. However the doctors have warned us about the unknown effects of combining Chinese and Western medical treatments and how it could be dangerous for my Dad if he was going through chemo, so we did not explore this area but stick with Western medicine."* 016M.

One participant, a licensed acupuncturist, shared an incident where she was denied immediate attention at the emergency room because her appearance looked "too well" for urgent care, despite her being covered with acupuncture needles while self-managing excruciating pain. She actually had a ruptured intestine, a life-threatening condition requiring emergency care. Despite being able to drive herself and even walk into the emergency room, the triage nurse misunderstood her situation:

*"The triage nurse wouldn't let me in so I walked over there and confronted her. Then she told me that everyone else was more urgent than I was, I was so furious as if my eyes were going to pop. . . I replied 'How do you know that I am not urgent?' Actually, my head, my hands and my legs were all covered with acupuncture needles. I couldn't wear shoes so I was in my slippers."*

—018F.

### 3.4.4. Cancer Stigma

Many participants were hesitant to discuss their symptoms and other cancer-related issues with friends or healthcare providers. Some attributed this reluctance to embarrassment or the stigma associated with cancer. Others believed that the effort to discuss would be futile. The majority simply tolerated their symptoms or addressed the problems as they arose:

*"So I asked them to take it (ostomy bag) off within 2 months. . . Back then I didn't always want to go outside. I didn't want to go to things like weddings and gatherings. . . I didn't want to tell others but now I'm open about it. If I tell others about this, they'd think that I deserve this."*

—001F.

They made significant adjustments to their social lifestyles to accommodate the side effects caused by the treatment:

*"Now when I go out, I won't eat anything. If anything were to happen, I wouldn't be eating outside. Rather, I'd eat at home because I can go to the washroom whenever I want, right? ...I'm always worried that I would need to buy bigger-sized t-shirts to make sure it covers my ostomy bag."*

—001F.

Participants felt that sharing their negative cancer experiences would invite insults and blame. Additionally, they viewed supportive services as unsuitable, non-essential, or inaccessible due to geographic, linguistic, and/or cultural barriers. Moreover, with limited knowledge about how the healthcare system worked, they chose to rely on themselves and settled for suboptimal disease management.

## 4. Discussion

The aim of this study was to explore the psychosocial and supportive care experiences of older Chinese immigrants living with cancer. A majority of the study participants either expressed a lack of need for supportive care or were unfamiliar with the limited available supportive services. Empathetic communication as the focal point of patient-centred care may be insufficient for this group of patients. The overarching theme of self-reliance as a virtue encapsulates how older Chinese individuals showed a subdued emotional response toward death and dying. This was attributed to a passive approach in care, emphasizing self-management of distress and symptoms to prevent burdening their family. Specifically, older Chinese adults view their life cycle as nearing completion, believing family resources should not be spent on psychosocial needs that could be managed independently. This perspective is exacerbated by language barriers, financial constraints, limited health literacy, stigma, and social barriers among these older individuals. Consequently, many of them endure their symptoms and settle for suboptimal disease management, potentially resulting in increased reliance on emergency care. To improve patient-centred care, our healthcare system needs to improve practical support, such as providing transportation services. This would directly alleviate the disease burden on patients' families.

The current findings challenge the common assumptions and focus on the vulnerabilities of older ethnic minority patients and argue that Chinese patients might be more resilient than the researchers and service providers thought. We observed that the motivation for self-management of psychosocial needs is shaped by their background and cultural perspective. This perception of a non-affective reactive approach to death and dying while emphasizing self-reliance to fulfill their social and familial roles is reinforced by submissive compliance to numerous barriers within the healthcare system. Older Chinese adults often seek to reduce the burden on their families by downplaying their need for supportive care, normalizing pain, and disregarding complications during the disease process [40]. However, depending solely on self-management for supportive care needs may yield negative consequences; prematurely relinquishing control over their health and treatment may result in worsening health conditions, ultimately necessitating more immediate healthcare resources. Further exploration is needed to determine if the inclination of internalizing psychosocial needs is associated with the desire to uphold a positive self-sacrificing image for their progeny, as indicated in previous research [41].

The sociocultural trait of self-reliance among older Chinese adults, aimed at reducing familial or societal burdens, creates a paradox in both relational and self-care aspects. This contradiction exists in the participant's pragmatic acceptance of death and dying, coupled with the desire to minimize familial burden and be self-reliant, which de-emphasizes their care needs. This neglect diminishes the acknowledgement of symptom severity, leading to inadequate symptom management. The overemphasis on self-reliance contributes to poorly managed symptoms, resulting in a decline in the patient's independence and an increased dependence on family and urgent healthcare resources. Whether older Chinese

adults are aware of this paradox or not warrants further research. In extreme cases, a desire for hastened death was evident. Two participants expressed hopelessness about their prognosis, considering refusal of treatment. One of them contemplated excessive drinking to expedite their death. Both participants had the misconception that refusing treatment would hasten their deaths and lessen their suffering. Their misconception was influenced by witnessing friends or loved ones who suffered tormenting treatments before passing away. There are no culturally appropriate terms in Chinese to articulate intentions like hastening death. Actively pursuing clinical trials was perceived as a selfish act, seen as a misuse of the healthcare system's resources by certain participants. While miscommunication and dissatisfaction with doctors may contribute to these perspectives [41], these patients were not aware that as the disease progresses, the greater severity of symptoms and complications could further compromise their quality of life without proper palliative care in place. Exploring potential explanations, age was considered a factor in the underutilization of health services. Upon closer investigation, it was found that age is not a determining factor, as there is an increased utilization of health services as individuals age [42–44]. Future patient education initiatives should concentrate on specific topics, such as treatment decisions, emphasizing independence, and fostering a positive family legacy in life with the philosophical and pragmatic views observed among older Chinese adults.

Consistent with other research, cultural, linguistic, and health literacy barriers affect the health-seeking behaviours of elderly Chinese immigrants [45,46]. Furthermore, there is a preference for cancer information from trusted sources such as family and friends. It may be worthwhile to further explore potential gender differences in attitudes and knowledge among this population. Without the ease of communication, barriers can include language difficulties and limited time with physicians, contributing to differing levels of health literacy. Our findings indicate that barriers to quality care extend beyond linguistic and literacy concerns. The cultural competence and confidence levels of the Chinese patient population toward the Canadian healthcare system might also contribute to the under-utilization of supportive care. Notably, some older Chinese adults felt that the Canadian healthcare system could not address their concerns regarding cancer treatment. At times, this clashed with their traditional Chinese medicinal practices. Some participants expressed frustration with lengthy diagnostic and presurgical tests, compounded by a lack of understanding about how the system operated. This was exacerbated by their knowledge and comparison with the healthcare system amenities in China. They also exhibited reservations in communicating their concerns on treatment, disease, and symptom management with their physicians. A participant had disregarded his Canadian doctor's advice and instead returned to China, seeking a more radical treatment based on perceived amenities. Overall, the findings suggest that certain older Chinese patients may perceive Canadian doctors to be less qualified compared to those in China, leading to a lack of trust in the healthcare system. It is noteworthy that trust in the Canadian healthcare system may vary based on a Chinese individual's immigration status and the duration of their stay in Canada [12]. Future studies could explore the perceived physician competence and trust issues between immigrant patients and healthcare providers. Leaders of healthcare institutions may consider additional cultural training, such as incorporating knowledge about traditional Chinese medicine practices for self-care. This could potentially enhance communication and cultural competency among staff members.

Given our findings, it is valuable to explore the perspectives of Chinese citizens and understand the disparities in healthcare attitudes between those in China and immigrants in Canada. China's collectivistic culture emphasizes social relationships and interdependence, influencing their response to trusted information. For both U.S. and Chinese citizens, similarities are shared with perceived cancer risk, positively correlating with cancer information-seeking behaviours [47]. However, there are differences in response to cancer worry with U.S. respondents being more proactive and less avoidant compared to Chinese respondents. The authors suggest that the Chinese may search for information that reduces their worry about cancer while avoiding information suggestive of a positive

cancer diagnosis [47]. Driven by cancer fatalism, Chinese immigrants may seek supportive care more actively when they perceive their adversities as concerns about cancer within the family.

Lastly, cancer fatalism is not exclusive to Chinese culture as it was observed in African American cancer patients [48,49]. Similarly, this study critically highlighted the view of cancer as a death sentence and found disparities regarding cancer knowledge and health behaviours. Additional studies have highlighted disproportionate fatalism among low-income minority populations affected by health disparities [49–51]. Most economically disadvantaged patients commonly hold the belief that death is inevitable when facing a serious disease [48]. Freeman proposed that poverty could be both a direct and an indirect influence on fatalism, possibly playing a role in factors such as undereducation, lack of healthcare access, and poor healthcare outcomes. The emphasis on day-to-day survival rather than preventive health measures like cancer screening, especially in the absence of symptoms, often results in late-stage cancer diagnosis and limited treatment options, leading to unfavourable outcomes. Beyond socioeconomic factors, collectivist thinking and the response to cancer-related uncertainties may contribute to fatalistic beliefs about cancer among older Chinese adults. Our findings suggest that Chinese immigrants, often misunderstood in prior research, exhibit resilience. Therefore, healthcare providers should adapt their communication style during care delivery, such as delivering distressing news, to achieve culturally competent care.

Limitations include the fact that the majority of participants were male, and while no significant gender-based differences in opinions were observed, the findings primarily apply to male older Chinese cancer patients. Upon examining gender differences, it was noted that GI cancer is more prevalent in males than females. This gender imbalance in sampling resulted from the limited availability of female Chinese cancer patients during recruitment and the lower prevalence of GI cancer among female patients. Future studies should allocate additional resources to balance the gender representation of subjects, especially for other chronic conditions. Furthermore, different stages or types of GI cancer may yield different effects of the illness. The diversity in experiences could have influenced their opinions and viewpoints. Finally, certain Chinese expressions had no direct translation into English, resulting in slight alterations in meaning. In addition, it may be beneficial to differentiate between subgroups within the Chinese minority, such as those from China and Hong Kong, as there could be variations in perception and belief systems. Given the limited sample size of interviews conducted, it is important to consider the shared perspectives of these individuals for insights. Future research may benefit from larger descriptive or correlational studies to further explore supportive care needs for elderly Chinese immigrants with GI cancer.

The COVID-19 pandemic has highlighted the necessity for culturally sensitive educational programs addressing patient attitudes and cultural factors contributing to the underutilization of health services. The current supportive care system largely follows a care-receiving model tailored for English-speaking populations, where patients are primarily positioned as the receivers of care. Understanding the specific needs of the ethnic minority patient population and evaluating the extent to which the needs are met is urgently required. Recent research suggests adopting a strength-based approach to promote self-care, emphasizing the choice and freedom of familial involvement [52]. The strength-based approach draws on the strengths and motivations of both the patient and their family to guide health-seeking behaviours. It underscores self-care and empowers patients to take an active role in their recovery by allowing them the autonomy to pursue health aligned with their individual values. A trained professional would conduct a detailed assessment to identify the personal strengths and resources of a patient while educating family members in the process. This approach is especially fitting considering that older Chinese patients prioritize their families. It is expected to enhance the patients' health-seeking behaviours and reduce their perceptions of burdening society. As the encouragement and promotion of diversity, equity, and inclusion (DEI) in healthcare services continue beyond the COVID-19

period [53], the importance of implementing culturally relevant services, notably in initiatives such as cancer screening programs, becomes more prominent. Recognizing the numerous barriers faced by older ethnic minorities, like Chinese seniors, encourages better structural design in the healthcare system. This ensures patients can navigate and embrace healthcare support services by overcoming the challenges associated with access and acceptance [54].

## 5. Conclusions

The lack of access to supportive care among older Chinese patients can be attributed to various factors encompassing cultural beliefs, individual barriers, systemic obstacles, and a philosophical yet pragmatic outlook on life and illness. While language challenges and limited understanding of healthcare play a role, the older Chinese view their life as nearing completion with no life-saving value. Therefore, supportive care that could burden family resources without significant benefits is discounted. They adopt a premature fatalistic acceptance of cancer, believing that altering the inevitable mortality is futile. This ingrained notion of life as predestined and natural sanctions the endurance of all cancer-related sufferings. Resultantly, older Chinese patients often internalize their needs and concerns, enduring them in silence. The many attitudes and feelings expressed by the participants in our study are not right or wrong but rather a personal decision regarding how one may decide to approach death. When designing educational resources for older Chinese immigrant patients, a family-centred approach is crucial. Utilizing a strength-based approach to identify their strengths, empowering both patients and their families, and advocating collective health management efforts could enhance behavioural and perspective changes. This approach may aid in diminishing health disparities among minority patients while decreasing the general misconception about their lack of participation in healthcare. Furthermore, this approach may promote openness and establish a positive attitude towards help-seeking behaviour, directly delaying acceptance of fatalism. With the emphasis on the family unit, an assessment would be helpful in identifying resources and establishing criteria for effective goal setting. This way, objective development is ensured and may allow for further revisitation in the future of expectations and overall acceptance.

**Author Contributions:** The following are individual contributions according to each author. Conceptualization, D.H. and E.W.Y.K.; methodology, E.W.Y.K. and Y.W.L.; software, E.W.Y.K. and Y.W.L.; validation, F.P., Y.W.L., and K.L.Y.W.; formal analysis, Y.W.L.; investigation, E.W.Y.K. and E.C.; resources, D.H. and A.M.; data curation, Y.W.L., E.W.Y.K., and F.P.; writing—original draft preparation, Y.W.L., K.L.Y.W., T.C., and J.S.; writing—review and editing, K.L.Y.W., J.S., and T.C.; supervision, D.H. and A.M.; project administration, Y.W.L.; funding acquisition, E.W.Y.K. and D.H. All authors have read and agreed to the published version of the manuscript.

**Funding:** This research was funded by the Union for International Cancer Control (UICC) Yamagiwa-Yoshida Memorial International Cancer Study Grant.

**Institutional Review Board Statement:** This study was conducted in accordance with the Declaration of Helsinki and approved by the Institutional Review Board of University Health Network (protocol code 14-7661 and date of approval 15 May 2014).

**Informed Consent Statement:** Informed consent was obtained from all subjects involved in the study.

**Data Availability Statement:** The data presented in this study are available on request from the corresponding author, Yvonne Leung.

**Acknowledgments:** We would like to thank all the patient participants. We would also acknowledge all research volunteers: Elrica Au, Niki Cai, Derrick Chan, Lindsay Chan, Clara Ho, Charles Lau, Melanie Le, Tiffany Lo, Sherry Song, Zhu Wang, Natalie Wong, Jack Zhang, and Vivian Zhou for their help in the literature review, transcription, translation, and editing.

**Conflicts of Interest:** The authors declare no conflicts of interest. The funders had no role in the design of the study; in the collection, analyses, or interpretation of the data; in the writing of the manuscript; or in the decision to publish the results.

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
