# Peer review of "Understanding Elderly Chinese Cancer Patients in a Multicultural Clinical Setting: Embracing Mortality and Addressing Misperceptions of Vulnerability"

_curroncol, doi:10.3390/curroncol31050197_

Round 1

Reviewer 1 Report

Comments and Suggestions for Authors

I believe the title should be specific (mention Chinese patients diagnosed with cancer) and include the specific outcome measure. The study did not include health care providers, hence the title and throughout the text should be amended.

Suggest presenting 3.1. Acceptance and Relinquishment in a more easy-to-follow approach (maybe expand over the page rather than try to present in a small space that is not easy to follow).

Author Response

Response to R1:

Title should be specific (mention Chinese patients diagnosed with cancer) and include the specific outcome measure. The study did not include health care providers, hence the title and throughout the text should be amended.

Response: The title was rewritten to improve accuracy and specificity of the subject matter. The sentences were amended (line 32 and line 74) to improve clarity and specificity. Other sentences containing “health care providers” involved participants’ thoughts or opinions and related to suggestions for improvements in healthcare services.

Suggest presenting 3.1. Acceptance and Relinquishment in a more easy-to- follow approach (maybe expand over the page rather than try to present in a small space that is not easy to follow).

Response: Thank you for the suggestion. We have elaborated more on the sub-theme of Acceptance and Relinquishment (line 171-175). The approach to elaborate on the sub-theme as been kept, keeping the structure consistent.

Reviewer 2 Report

Comments and Suggestions for Authors

Dear authors,

Many thanks for submitting your work to the journal. I have now uploaded my evaluation as an attached file.

Best regards

Author Response

Response to R2:

Manuscript’s discussion section would benefit from a more thorough comparative analysis of the study findings and a more detailed discussion of the study’s limitations. The manuscript could benefit from a more detailed discussion on how these limitations impact the study findings and suggestions for future research. Also providing a discussion on the generalizability of the findings beyond the study population would strengthen the manuscript.

Response: Thank you for suggesting a more thorough comparative analysis of study findings and limitations.  We have added additional considerations in similarities and differences with other similar studies within the discussion (line 545, 573, 589, and 612). Regarding the discussion of the generalizability, qualitative studies are usually for diving into rich contexts to uncover nuances and complexities of smaller samples and for understanding intricates of human experiences. This allows for themes and patterns to emerge.

The conclusion section needs a thorough exploration of the practical implications of the study findings for healthcare practice, policy development, and future research. Making concrete recommendations based on the study results would improve the research’s relevance and applicability. Make proposals for future research based on findings of the present study. Provide clear recommendations for future research to guide scholars in addressing gaps identified in current study.

Response: Thank you for the suggestion to guide future research in addressing gaps of our study. We have included a few direction for future research in our discussion/ limitations. It may be redundant to include in the conclusion as well. Regarding the practical implications of the study, due to limited generalization, we have instead suggested a strength-based approach with a framework of what can be applied (line 654).

Reviewer 3 Report

Comments and Suggestions for Authors

The authors describe an interesting and important examination of Chinese immigrants' perceptions about their treatment for GI cancer. The qualitative design is appropriate for capturing these attitudes in a personal manner and the findings are well presented in participant quotes. The results also appear to be appropriately interpreted.

Comments:

Methods

Line 111: clarify that the 10 research volunteers were research staff. "Volunteer" implies participant.

When was the study conducted?

Table 1: Suggest a table format with at least upper and lower boundary lines.

Figure 1: Title/legend should be located below the figure. Spelling needs correction. Clarify the meaning of the different types of dotted lines around the themes/subthemes.

Results: It is common to provide a brief description of the speaker of the quote, such as (male, age 70 y, with stage IV pancreatic cancer). Include even a minimal description, if possible, without violating the privacy of the patient.  

Discussion: A description of how terminal disease is handled by Chinese citizens would be valuable. It would help the reader understand the differences between health care attitudes by those in China vs. immigrants in Canada.

I sense from the authors a general disapproval of many of the attitudes toward terminal disease expressed by the participants. It may be appropriate to state that such feelings are not "right" or "wrong" but rather a personal decision/individual agency regarding how one decides to approach death.

Comments on the Quality of English Language

Minor English edits are needed including the title: The Misperception of Health Care Providers on of Non-English 2 Speaking Cancer Patients’ Vulnerability in a Multicultural 3 Clinical Setting.

Author Response

Response to R3:

Methods Line 111: clarify that the 10 research volunteers were research staff. “Volunteer” implies participant.

Response: Thank you for pointing that out. We have edited it by changing to “research volunteer staff”.

When was the study conducted?

Response: Thank you for pointing that out. We have added in the study dates in line 118-119.

Table 1: Suggest a table format with at least upper and lower boundary lines.

Response: Thank you for the suggestion. We have added boundary lines surrounding the whole table.

Figure 1:  Title/ legend should be located below the figure. Spelling needs correction, Clarify the meaning of the different dotted lines around the themes/ subthemes.

Response: Thank you for pointing that out. We have placed the titles below the figures and have corrected the spelling error for the title. We have also included a note to clarifying the borders corresponding to each grouping.

Results: It is common to provide a brief description of the speaker of the quote. Include even a minimal description, if possible, without violating the privacy of the patient.

Response: Thank you for this suggestion. We have added the participant ID# and gender at the end of each quote without disclosing any descriptors that may violate the privacy of the patient.

Discussion: A description of how terminal disease is handled by Chinese citizens would be valuable, It would help the reader understand the differences between health care attitudes by those in China vs. immigrants in Canada.

Response: It is a very relevant perspective. We do not have an in-depth understanding of differences between healthcare attitudes and should require more investigation. In the discussion, we have included a possible view to consider when looking at Chinese citizens and immigrants in Canada (line 573).

I sense from the authors a general disapproval of many of the attitudes toward terminal disease expressed by the participants. It may be appropriate to state that such feelings are not “right” or “wrong” but rather a personal decision/ individual agency regarding how one decides to approach death.

Response: We are not necessarily disapproving of the attitudes, but more so wanting to shed light on their perspectives’. Where some may find their views to be unreasonable, we see these Chinese immigrants as resilient and unique. To clarify this, we have amended this perspective in the conclusion (line 652).

Round 2

Reviewer 3 Report

Comments and Suggestions for Authors

The authors have addressed my concerns adequately.